# Genome-Wide Association Study for Detecting Salt-Tolerance Loci and Candidate Genes in Rice

San Mar Lar [1], Jeonghwan Seo [1,2], Seong-Gyu Jang [1], Hongjia Zhang [1], Ah-Rim Lee [1], Fang-Yuan Cao [1], Ja-Hong Lee [1], Na-Eun Kim [1], Yoonjung Lee [3], Yong-Jin Park [4], Joohyun Lee [3,*] and Soon-Wook Kwon [1,2,*]

1 Department of Plant Bioscience, College of Natural Resources and Life Science, Pusan National University, Miryang 50463, Korea; sanmarlar2010@gmail.com (S.M.L.); rightseo83@gmail.com (J.S.); sgjang0136@gmail.com (S.-G.J.); hjzhangpnuedu@gmail.com (H.Z.); aar5430@gmail.com (A.-R.L.); fangyuan3507@gmail.com (F.-Y.C.); jhlp0921@gmail.com (J.-H.L.); rlaskdms18@gmail.com (N.-E.K.)
2 Life and Industry Convergence Research Institute, Pusan National University, Miryang 50463, Korea
3 Department of Crop Science, Konkuk University, Seoul 05029, Korea; yoon10.lee@gmail.com
4 Department of Plant Resources, College of Industrial Sciences, Kongju National University, Yesan 32439, Korea; yjpark@kongju.ac.kr
* Correspondence: edmund@konkuk.ac.kr (J.L.); swkwon@pusan.ac.kr (S.-W.K.); Tel.: +82-2-450-3769 (J.L.); +82-55-350-5506 (S.-W.K.)

**Abstract:** Salinity is one of the major constraints causing soil problems and is considered a limitation to increased rice production in rice-growing countries. This genome-wide association study (GWAS) experiment was conducted to understand the genetic basis of salt tolerance at the seedling stage in Korean rice. After 10 days of salt stress treatment, salt tolerance was evaluated with a standard evaluation system using a visual salt injury score. With 191 Korean landrace accessions and their genotypes, including 266,040 single-nucleotide polymorphisms (SNPs), using a KNU Axiom Oryza 580K Genotyping Array, GWAS was conducted to detect three QTLs with significant SNPs with a $-\log10(P)$ threshold of $\geq 3.66$. The QTL of *qSIS2*, showed $-\log10(P) = 3.80$ and the lead SNP explained 7.87% of total phenotypic variation. The QTL of *qSIS4*, showed $-\log10(P) = 4.05$ and the lead SNP explained 10.53% of total phenotypic variation. The QTL of *qSIS8* showed $-\log10(P) = 3.78$ and the lead SNP explained 7.83% of total phenotypic variation. Among the annotated genes located in these three QTL regions, five genes were selected as candidates (*Os04g0481600*, *Os04g0485300*, *Os04g0493000*, *Os04g0493300*, and *Os08g0390200*) for salt tolerance in rice seedlings based on the gene expression database and their previously known functions.

**Keywords:** GWAS; salt-tolerance; Korean landrace rice; candidate gene

---

## 1. Introduction

Rice is grown in more than 100 countries over approximately 158 million hectares, with production of more than 700 million tons annually. Yields range from less than 1 t/ha under very poor rainfed conditions to more than 10 t/ha in intensively irrigated systems in temperate conditions (http://ricepedia.org/rice-as-a-crop/rice-productivity, accessed on 11 September 2021). Salinity is one of the major constraints causing soil problems and is considered as a limitation to increased rice production in rice-growing countries.

Salinity includes all soil problems due to excessive salt; these soils are categorized as sodic (or alkaline) and saline soils. Sodic soils can occur widely in arid and semi-arid regions, and excessive $Na^+$ occurs at the exchangeable sites of clay particles. These soils have higher than pH 8.5 and a high exchangeable sodium percentage (ESP > 15). Saline soils are generally distributed in arid regions, estuaries, and coastal fringes and are dominated by $Na^+$ cations with electrical conductivity (EC) > 4 dSm$^{-1}$. In saline soils, ESP values are <15 and pH values are much lower than those of sodic soils (http://www.knowledgebank.irri.org/ricebreedingcourse/Breeding_for_salt_tolerance.htm, accessed on 11 September 2021). Generally, rice is relatively tolerant to stress during germination, active tillering, and

maturity, but is particularly sensitive at the both the seedling and reproductive stages [1]. Rice is included among the most sensitive cereal crops, with a threshold level of 3 $dSm^{-1}$. Moderate salinity can be defined as electrical conductivity of the soil or solution culture of 4 $dSm^{-1}$, and high salinity as 4–8 and 8–12 $dSm^{-1}$ [2].

Symptoms of salinity stress at the seedling stage include whitened tops of affected leaves, chlorotic patches on some leaves, plant stunting, reduced tillering, and patch field growth, which can lead to death [3]. Major symptoms of salinity at the reproductive stages are white leaf tip followed by tip burning, stunted plant growth, low tillering, spikelet sterility, low harvest index, fewer florets per panicle, less 1000 grain weight, low grain yield, change in flowering duration, leaf rolling, white leaf blotches, poor root growth, and patchy growth in the field [4]. A 12% reduction in yield will occur with every unit ($dSm^{-1}$) increase in electrical conductivity above the threshold level [5–8]. Salt concentrations of 4, 8, and 12 $dSm^{-1}$ were reported to lead to significant grain yield reduction of 31, 56, and 71%, respectively [9]. Hasamuzzaman et al. [10] reported 36.17–50% grain loss due 150 mM salinity. Another study showed that when rice seedlings were inhibited by salt, the shoot length and dry root weight were significantly decreased, but chlorophyll content, fresh shoot weight, and dry shoot weight were increased [11].

Identifying QTLs related to salt-tolerance traits has become a major effort for breeding programs. By using targeted marker-assisted backcrossing and selection, several salt-tolerant rice lines have been developed by introgression of Saltol QTL (major QTL of seedling stage tolerance) in salt-sensitive, high-yielding varieties [12,13]. Many scientists have carried out experiments and detected QTLs related to salt tolerance. In $BC_3F_4$ introgression lines of Pokkali developed in IR29, 6 QTLs for salt injury score (SIS) were associated with chromosomes 1, 3, 4, 10, and 11 [14]. Seven QTLs were identified for SIS on chromosomes 2, 5, 6, 7, 8, 9, and 11 in a Bengal/Pokkali RIL population [15], and one major QTL on chromosome 5 in Nona Bokra IL lines [16]. Two major QTLs for shoot $Na^+$ and $K^+$ concentration were found on chromosomes 7 and 1 [17]. Rahman et al. [18] identified three QTLs related to salt tolerance at the seedling stage on chromosome 1 and four QTLs on chromosome 3 that were located at 162.9 cM and 111 cM, respectively.

Genome-wide association study (GWAS) has great potential for identifying valuable natural variations in trait-associated loci, as well as allelic variations in candidate genes underlying quantitative and complex traits, including those related to growth, development, stress tolerance, and nutritional quality [19]. Naveed et al. [20] identified 6 and 14 QTLs for salt-tolerance traits at the germination and seedling stages in 208 rice accessions. Another study demonstrated a significant association of 10 genes with salt tolerance related traits at seedling and yield stages and related traits in rice plants grown under saline conditions [21]. A total of 11 QTLs were identified at the germination and early seeding stages in *japonica* rice [22]. Yu et al. [23] obtained 93 candidate genes significantly associated with salt tolerance in rice at the seedling stage.

Hoang et al. [24] generated transgenic rice expressing the anti-apoptotic genes *AtBAG4*, *Hsp70*, and *p35* for enhanced salinity tolerance with many characteristics (maintenance of shoot growth, dry weight, number of panicles, number of spikelets; suppression of programmed cell death (PCD) pathway; detoxification of reactive oxygen species (ROS); minimization of cellular membrane electrolyte leakage; high photosynthetic efficiency; low $Na^+$ accumulation). Transgenic rice harboring *HsCBL8* gene was isolated from XZ166, a wild-type barley line. *HsCBL8* gene encodes a calcium-sensor calcineurine B-like (CBL) protein in rice, and its overexpression leads to significant improvement in water protection and plasma membrane in vivo, more proline accumulation, and reduced overall $Na^+$ uptake, but little change in $K^+$ concentration in the plant tissues [25]. Tang et al. [26] studied the overexpression of *OsMYB6* gene in transgenic rice lines at the seedling stage for salinity stress. After 6 days in salt solution, *OsMYB6* transgenic rice plants resulted in more green leaves and less leaf wilting and rolling than the wild-type plant. When the seedlings were moved to Yoshida solution, all wild-type seedlings were dead and 43.9% transgenic seedlings survived.

Sensing and signaling (stress and ROS scavenging and signaling) and functional adaptation, including stomatal regulation, osmotic adjustment, and ion homeostasis, are the main factors of salt-tolerant molecular mechanisms [27]. Overexpression of *OsMYB91* gene enhanced salt tolerance, increased ROS scavenging ability, and increased proline levels [28]. The salt- and drought-inducible ring-finger 1 (*OsSDIR1*) gene was upregulated under drought and high salinity, and its overexpression resulted in enhanced tolerance to water deficit in plants by decreasing water loss, mediated by stomatal closure [29]. Functional analysis of *SKC1* gene shows that it encodes an HKT-type transporter (*OsHKTS1;5*), which is preferentially expressed in parenchyma cells surrounding the xylem vessels in rice and regulates $K^+/Na^+$ homeostasis under salt stress [30].

## 2. Materials and Methods

### 2.1. Plant Materials

In this study, 191 Korean landrace accessions were used to evaluate salt tolerance at the seedling stage (Table S1). Seeds of all accessions were acquired from the Rural Development Administration (RDA) Genebank, Jeonju, Republic of Korea (http://genebank.rda.go.kr, accessed on 5 July 2020).

### 2.2. Evaluation of Salt Tolerance

The seeds were heat treated at 50 °C for 3 days to break dormancy. To control microbial contamination and promote germination, the seeds were surface sterilized by soaking in sodium hypochlorite (NaClO) for 20 min and then washed with distilled water 3 times. Sterilized seeds were placed in Petri dishes lined with moistened filter paper and incubated at 30 °C for 3 days to germinate. The germinated seeds were sown in 96-well PCR plates, the bottoms of which had been cut. The PCR plates were suspended on distilled water, and after 2 days were transferred to nutrient solution (Yoshida solution) [31]. The salt screening was done in a controlled growth chamber (Hanbaek Sci. Bucheon, South Korea) at 29 °C/ 21 °C day/night temperature and 70% relative humidity. After 2 weeks in nutrient solution, the seedlings for salt treatment were introduced to salt stress by adding sodium chloride (NaCl) to the nutrient solution up to 150 mM. Before adding NaCl, 5 uniform plants per line were selected for salt screening and the others were removed. The pH of the nutrient solution was adjusted to 5.0–5.5 with hydrochloric acid (HCL) or sodium hydroxide (NaOH), and the solution was renewed every week. After 10 days in salt stress, the seedlings were evaluated for salt injury score by phenotype according to the standard evaluation system (Figure 1, Table 1) [32].

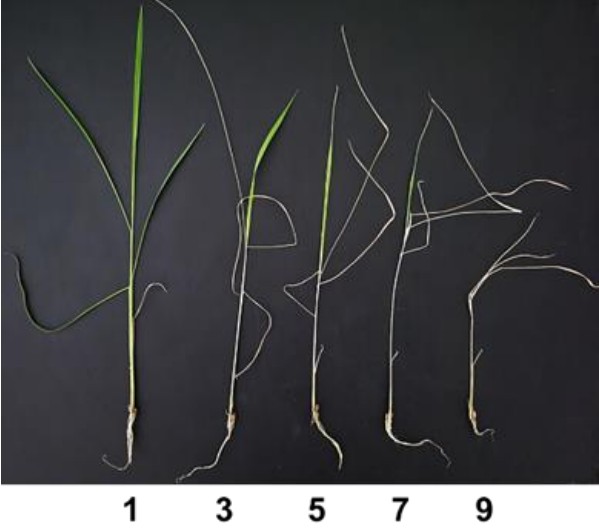

**Figure 1.** Salt injury score at seedling stage.

**Table 1.** Standard evaluation system of visual salt injury at seedling stage [32].

| Score | Observation | Tolerance |
|:---:|:---:|:---:|
| 1 | Growth and tillering nearly normal | Highly tolerant |
| 3 | Growth nearly normal but some reduction in tillering and some whitish and rolled leaves | Tolerant |
| 5 | Growth and tillering reduced, most leaves whitish and rolled, only a few elongated | Moderately tolerant |
| 7 | Growth completely ceased, most leaves dry, some plants dying | Susceptible |
| 9 | Almost all plants dead or dying | Highly susceptible |

*2.3. DNA Extraction and High-Throughput SNP Genotyping*

The genomic DNA of 191 landraces was extracted from fresh leaves of 14-day-old seedlings using the CTAB method [33]. The quality of DNA was checked by agarose gel electrophoresis and quantified using a Nanodrop ND-1000 spectrophotometer (Thermo Fisher Scientific, Wilmington, DE, USA). High-throughput SNP genotyping and genotype calling of 191 landraces were carried out using KNU Axiom Oryza 580K Genotyping Array and Affymetrix Power Tools according to the description in a previous study [34].

*2.4. Population Structure Analysis*

For population structure analysis in present populations, we first filtered the genotype data by PLINK software [35], then high-quality, low-LD SNP sets were collected and underwent subsequent analysis; PCA analysis and plot visualization were performed using R package (version 4.03, http://r-project.org, accessed on 7 June 2021) [36], and for structural analysis we used ADMIXTURE software with bed format file. Finally, the result was visualized by Pophelper web tools [37], and the delta k values were shown for identifying the population groupings.

*2.5. Genome-Wide Association Study (GWAS) Analysis*

The Trait Analysis by Association, Evolution, and Linkage (TASSEL) package [38] was used to conduct association analysis of the salt tolerance of 191 landrace accessions. The mixed linear model (MLM) was performed, in which a kinship (K) matrix as the variance–covariance matrix between individuals was combined with population structure from PCA. Due to the fact that many SNPs have strong LD in genotype data, the thresholds decided by the total number of SNPs were too rigorous for detecting association loci [39], thus the genotype was filtered by PLINK software [35]. Non-independent SNPs were removed, and a total of 4556 effective and independent SNPs remained, and the association threshold was calculated by the formula: $-\log 10(1/\text{number of independent SNPs})$ [40]. Finally, the threshold was set as $-\log_{10}(P) = 3.66$ for identification of association loci, and SNP markers located at locus peaks were designated as lead SNPs for the detected loci. The areas 300 kb upstream and downstream of the lead SNPs were considered as candidate genomic regions for gene identification.

*2.6. Identifying Candidate Genes for Salt Tolerance*

GWAS analysis was used to identify promising candidate genes for salt tolerance in rice. We identified candidate genes associated from 300 kb upstream and downstream of significant SNP regions. For the expression patterns of the genes located in QTL regions, the RNA-seq data of GSE119720 GEO accessions was obtained from the NCBI GEO database (https://www.ncbi.nlm.nih.gov/geo/, accessed on 11 September 2021) [41]. A heatmap was plotted by http://www.bioinformatics.com.cn, 13 September 2021, an online platform for data analysis and visualization.

### 2.7. Haplotype Analysis

The previously reported phenotype dataset for the salt stress study by Yu et al. [23] was used in the haplotype analysis. In the study of Yu et al., the phenotypes of the leaf width (LW) and the length of plant shoots (SL) and roots (RL) were measured immediately after salt treatment. The total dry weight (TDW) was measured after 80 °C for 24 h incubation. Relative TDW (R-TDW), relative shoot length (R-SL), and relative leaf width (R-LW) were measured as the ratio of TDW, SL, and LW under salt stress to those under control conditions. Haplotype variation analysis was performed using PopART software [42]. LD blocks within 300 kb upstream and downstream of significant SNP regions were constructed using HaploView 4.2 [43]. Visualization of gene structures and SNP positions of candidate genes was illustrated by the Gene Structure Display Server 2.0 online tool [44]. One-way analysis of variance followed by Duncan's test was used to compare phenotypic differences among haplotypes by SPSS version 26.0 (IBM Corp., Armonk, NY, USA).

## 3. Results

### 3.1. Salt Stress Tolerance at Seedling Stage

After 10 days in salt stress, salt tolerance in the seedlings was evaluated by phenotype according to the standard evaluation system. The distribution of salt injury scores was normal, with a score of 7 showing the highest frequency among 45 accessions (Figure 2). The average salt injury score of 191 accessions was 5.7. Accession ja110-Jwiiparibyeo (*japonica*) was the most salt-susceptible, with a score of 9.0, and the most tolerant accession was ja046-Jangsamdo (*japonica*), with a score of 1.67, followed by ja036-Noinjo (*japonica*), ja338-Duchungjong (*japonica*), and ja 281-Gangreungdo (*japonica*), with a score of 2.0.

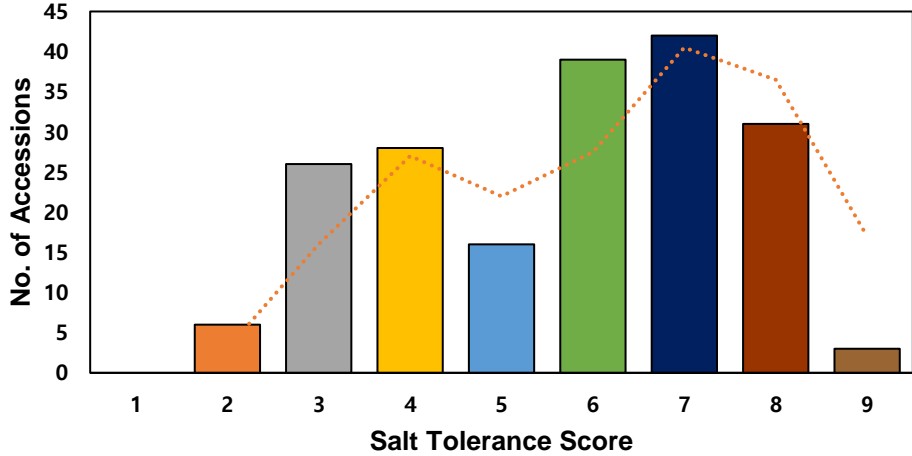

**Figure 2.** Frequency distribution of salt tolerance in 191 rice accessions. Histogram of salt tolerance score. Dotted line shows moving average (2 intervals). Colors indicate average salt tolerance score: orange, 2; gray, 3; yellow, 4; light blue, 5; green, 6; dark blue, 7; brown, 8; gold, 9.

### 3.2. Genetic Structure of 191 Rice Accessions

According to the PCA results (Figure 3A), 191 accessions were clearly separated into three clusters following the X- and *Y*-axis. PC1 matrix explained about 57% and PC2 matrix explained about 25% of phenotype variation in this study, indicating that PCA = 3 is a suitable grouping to reduce the population error in the present population. In structural analysis, large error was observed in K = 1 and K = 2, but it rapidly decreased when K = 3 (Figure 3B), indicating that the 3 is the most suitable group for this population, and in the detailed grouping results shown in Figure 3C, clusters 1, 2, and 3 represent the different groupings.

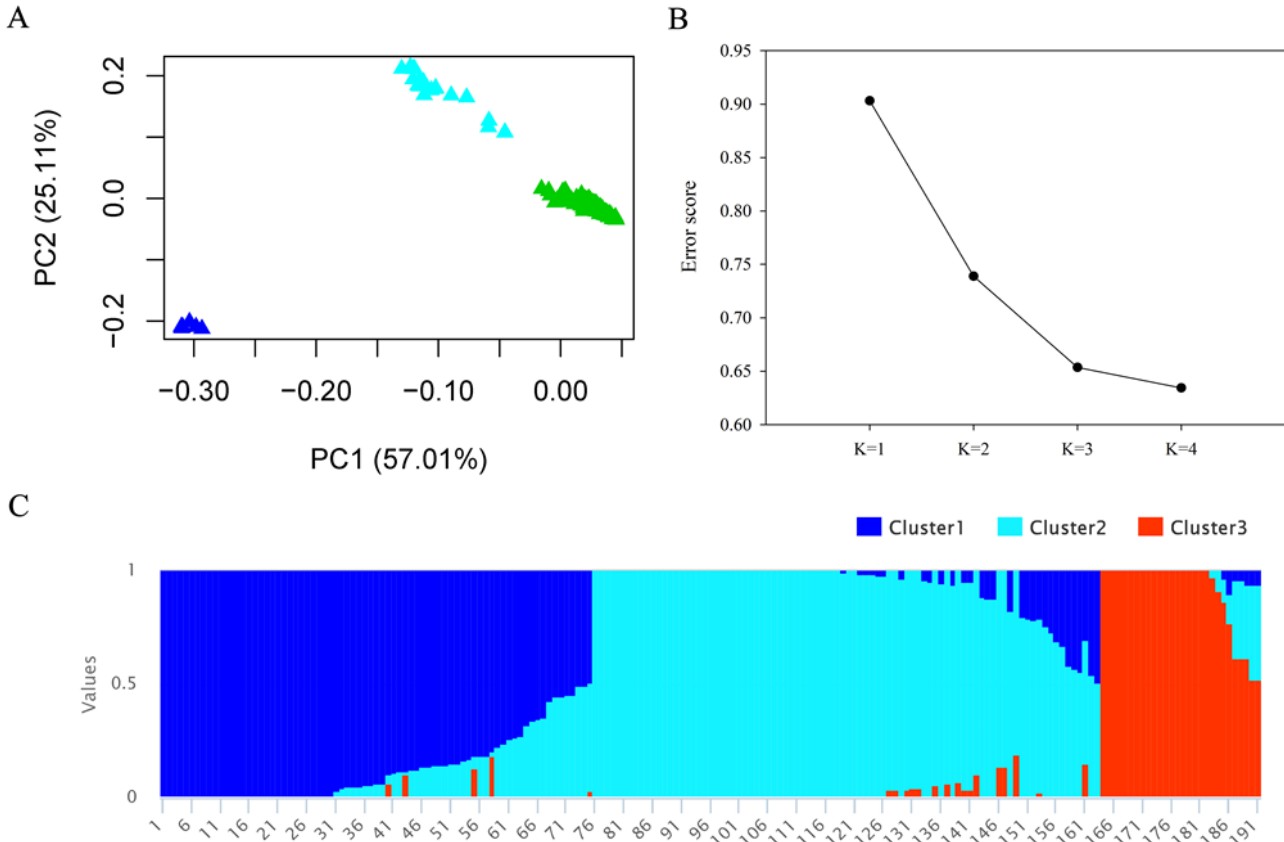

**Figure 3.** Population structure analysis of 191 rice accessions: (**A**) principal component analysis (PCA) (PC1 and PC2); (**B**) cross-validation (CV) error of diverse groups (K); (**C**) structural analysis outcome (K = 3).

### 3.3. GWAS Analysis

Genotyping was done with 266,040 single-nucleotide polymorphisms (SNPs) using KNU Axiom Oryza 580K Genotyping Array. The SNP dataset was filtered by using TASSEL version 5.2.33 [38] and PLINK software [35]. Minor allele frequencies (<0.03) and missing values (0.01) were removed. After filtering the genotype with minor allele frequency 0.03, 124,347 SNPs were left to analyze. The Manhattan plot for the markers significantly associated with salt tolerance at seedling stage is represented in Figure 4. Associations higher than the threshold of $-\log10(P) \geq 3.66$ were detected as significant SNPs (refer to Materials and Method). Significant SNPs within the 600 kb surrounding the lead SNPs were considered as one association locus, and, in total, three QTLs were mapped (Table 2).

One associated QTL was detected on chromosome 2, designated as *qSIS2*, showing $-\log10(P) = 3.80$, and the lead SNP explained 7.87% of total phenotypic variation. One associated QTL was detected on chromosome 4, designated as *qSIS4*, showing $-\log10(P) = 4.05$, and the lead SNP explained 10.53% of total phenotypic variation. The other associated QTL was detected on chromosome 8, designated as *qSIS8*, showing $-\log10(P) = 3.78$, and the lead SNP explained 7.83% of total phenotypic variation. For the regions of the three QTLs, previously reported QTLs were used for comparison (Table 2). There were no salt-tolerance-associated QTLs reported based on Gramene (http://archive.gramene.org, accessed on 21 September 2021). We detected only two reported QTLs associated with root response to soil environment. For the region of *qSIS4*, phosphorus-sensitivity-associated QTL was reported, and for the region of *qSIS8*, iron-sensitivity-associated QTL was reported.

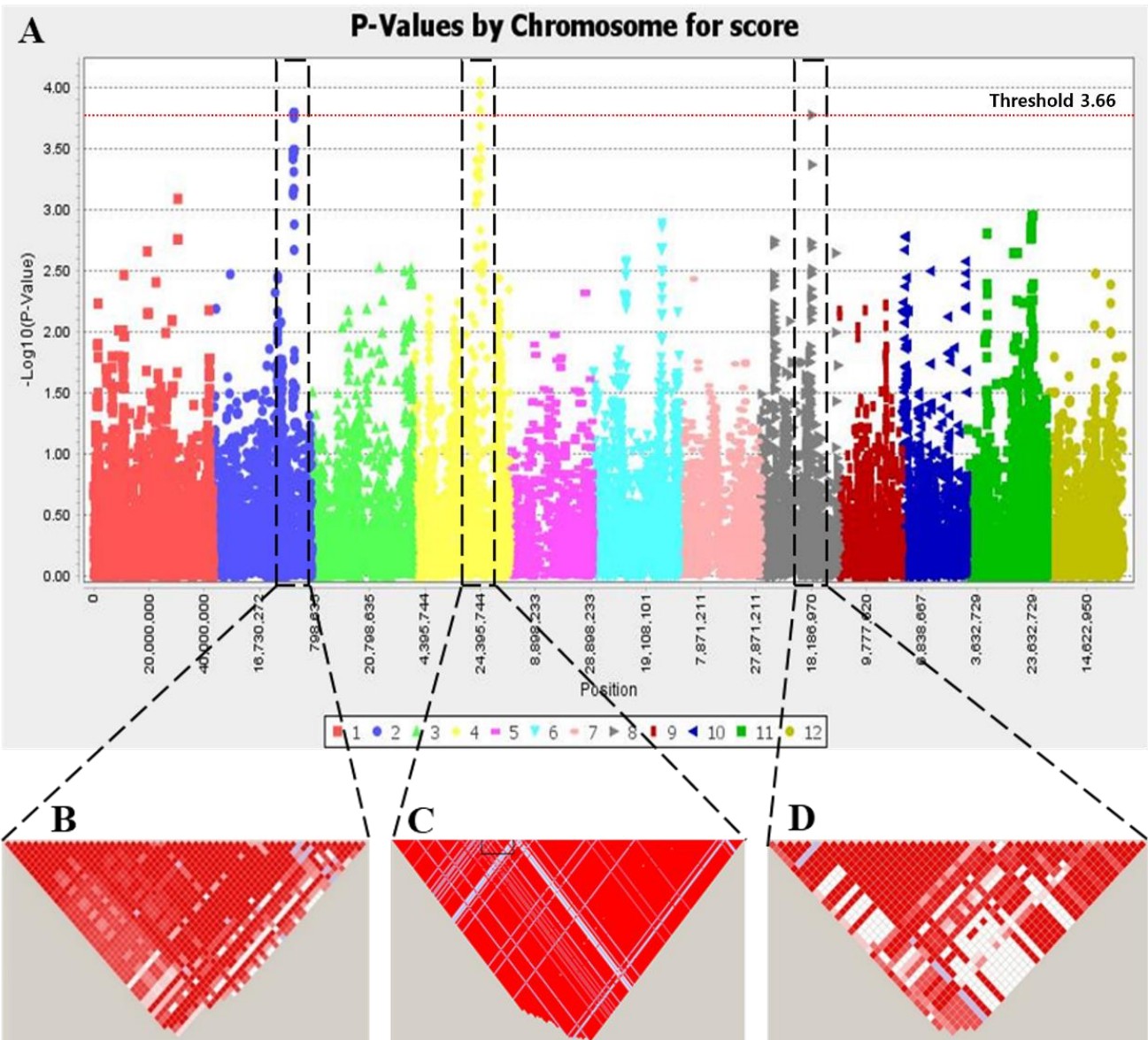

**Figure 4.** Genome-wide association mapping and LD block of salt injury score (SIS) under salt stress (150 mM). (**A**) Manhattan plot of genome-wide association mapping using MLM. Colors indicate rice chromosome: red, chr1; blue, chr2; yellow green, chr3; yellow, chr4; pink, chr5; light blue, chr6; crimson, chr7; gray, chr8; dark brown, chr9; dark blue, chr10; green, chr11; gold, chr12. Red-dotted line indicate threshold 3.66 (**B**–**D**) LD blocks for region of *qSIS2* on chromosome 2, region of *qSIS4* on chromosome 4, and region of *qSIS8* on chromosome 8, respectively.

**Table 2.** Locations of QTLs detected in GWAS and previously reported QTLs.

| QTLs | Chr. | Position of Lead SNP | $-\log_{10}(P)$ | Reported QTLs [1] | |
|---|---|---|---|---|---|
| | | | | QTL Accession | Reported Trait |
| *qSIS2* | 2 | 29138395 | 3.80 | - | - |
| *qSIS4* | 4 | 24390487 | 4.05 | AQCI011 | Phosphorus sensitivity |
| *qSIS8* | 8 | 18649847 | 3.78 | AQDP005 | Iron sensitivity |

[1] Gramene (http://archive.gramene.org, accessed on 20 September 2021).

### 3.4. Identifying Candidate Genes

To identify candidate genes responsible for salt tolerance, all annotated genes located within 600 kb of the QTL regions were extracted based on the RAP-DB (IRGSP 1.0). A total of 212 genes were located in the three QTL regions (Table S2): 83 genes in chromosome 2, 93 genes in chromosome 4, and 36 genes in chromosome 8. The expression patterns of 212 genes were searched with previously reported RNA-seq in the database [41]. The results revealed that 44 genes were significantly differentially expressed under salt-stressed conditions (Figure 5, Table S3).

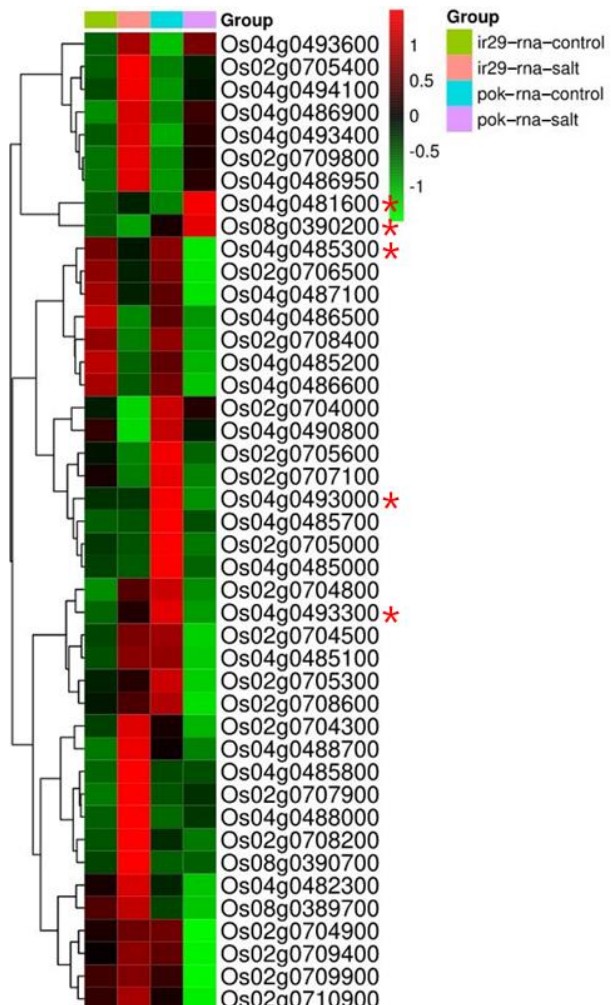

**Figure 5.** Gene expression analysis of 44 candidate genes related to salt stress between IR 29 and Pokkali. Red and green represent high and low expression level, respectively. Red asterisk mark (*) represents the candidate genes used in the haplotype analysis.

Among the 44 differentially expressed genes, 19 genes were increased under salt stress conditions. In the susceptible IR29 variety under salt-treated conditions, 12 genes were increased; these genes were not increased in the tolerant Pokkali variety. Five genes were increased in both susceptible and tolerant varieties, and two genes were increased only in the tolerant variety. Based on the expression patterns in the database, we selected *Os04g0481600* encoding WD domain, G-beta repeat domain containing protein, and *Os08g0390200* encoding B-box zinc finger family protein as the candidate genes associated with salt tolerance, because they were significantly increased in the salt-tolerant Pokkali variety under salt-treated conditions, but were not increased in the susceptible IR29 variety. In addition, three candidate genes were selected based on their association with salt toler-

ance previously reported in other plants such as soybean and Arabidopsis. *Os04g0485300* encodes glucose-6-phosphate 1-dehydrogenase, which was previously reported to have a possible role in salt tolerance in soybean [45] and reed [46]. *Os04g0493000* encodes B-box zinc finger family protein and *Os04g0493300* encodes glycine-rich protein. A possible role for these genes against salt tolerance in Arabidopsis was suggested [47,48].

### 3.5. Haplotype Analysis of Candidate Genes

For the five candidate genes selected based on the previously reported gene expression pattern and their reported function, we conducted haplotype analysis. Since our experiments were conducted with gene-chip based technology, we used a previously reported database constructed with a resequencing method to capture as many possible haplotypes as we could. For the phenotype and genotype data, the reported data by Yu et al. [23] for salt stress were used. In the previous study, 295 rice germplasms and genomic sequencing data were used for GWAS analysis for root and shoot growth traits under the 200 mM NaCl salt condition. Haplotype analysis of these five candidate genes is presented in Figures 6–10. Heterozygous SNPs and missing data were excluded, and SNPs from exons were used for haplotype and haplotype variation analysis. We detected significant differences among haplotypes varying in the five candidate genes for at least three out of the seven phenotypic traits (root length, shoot length, leaf width, total dry weight, relative shoot length, relative leaf width, and relative total dry weight). Based on the evaluated traits by Yu et al., the traits of relative shoot length, relative leaf width, and relative total dry weight can explain the salt tolerance by eliminating the intrinsic differences among accessions. Except *Os04g0493000*, we detected significant differences among haplotypes varying in the other four candidate genes for the trait of relative leaf width. There were seven non-synonymous SNP regions in *Os04g0481600* (WD domain, G-beta repeat domain containing protein) (Figure 6A). Among them, five SNPs showed amino acid substitution when alleles changed (C→T, Chr4_24087653, R→C substitution; T→C, Chr_24087785, C→R substitution; C→G, Chr4_2408783, H→D substitution; A→G, Chr4_24087804, H→R substitution; A→G, Chr4_24087809, T→A substitution). *Os04g0481600* consists of four haplotype groups (Figure 6B). Significant differences among haplotypes varying in *Os04g0481600* for root length, shoot length, leaf width, and relative leaf width were detected. For example, the relative leaf width of Hap1 and Hap2 of *Os04g0481600* differed significantly from Hap3. *Indica* varieties were assigned into Hap1, Hap3, or Hap4, while *japonica* varieties were assigned into Hap2 (Figure 6C).

The gene structure of *Os04g0485300* encoding glucose-6-phosphate 1-dehydrogenase is represented in Figure 7A. There are 15 exons and one SNP in exon 11 (C→T, Chr4_24265926, V→V), and one SNP in exon 15 (A→G, Chr4_24267469, E→E) showed changed allele without amino acid substitution. *Os04g0485300* consists of two haplotype groups (Figure 7B). Significant differences among haplotypes varying in *Os04g0481600* for root length, shoot length, leaf width, total dry weight and relative leaf width were detected. For example, the relative leaf width of Hap1 of *Os04g0485300* differed significantly from Hap2. Hap 1 includes only *indica* varieties, and Hap 2 consists of *indica* and *japonica* varieties (Figure 7C).

*Os04g0493000* encodes B-box zinc finger family protein. (Figure 8A). It has two exons, and one SNP in exon 1 (T→C, Chr4_24648785, T→A substitution), and allele A changed to C in exon 2 on position Chr4_24648070. *Os04g0485300* consists of two haplotype groups, and *Os04g0481600* consists of four haplotype groups (Figure 8B). Significant differences among haplotypes varying in *Os04g0493000* for root length, leaf width and relative leaf width were detected. The relative leaf width of Hap2 of *Os04g0493000* differed significantly from Hap1. The haplotype of Hap1 consists of *japonica* varieties, and the haplotype of Hap2 consists of *indica* (Figure 8C).

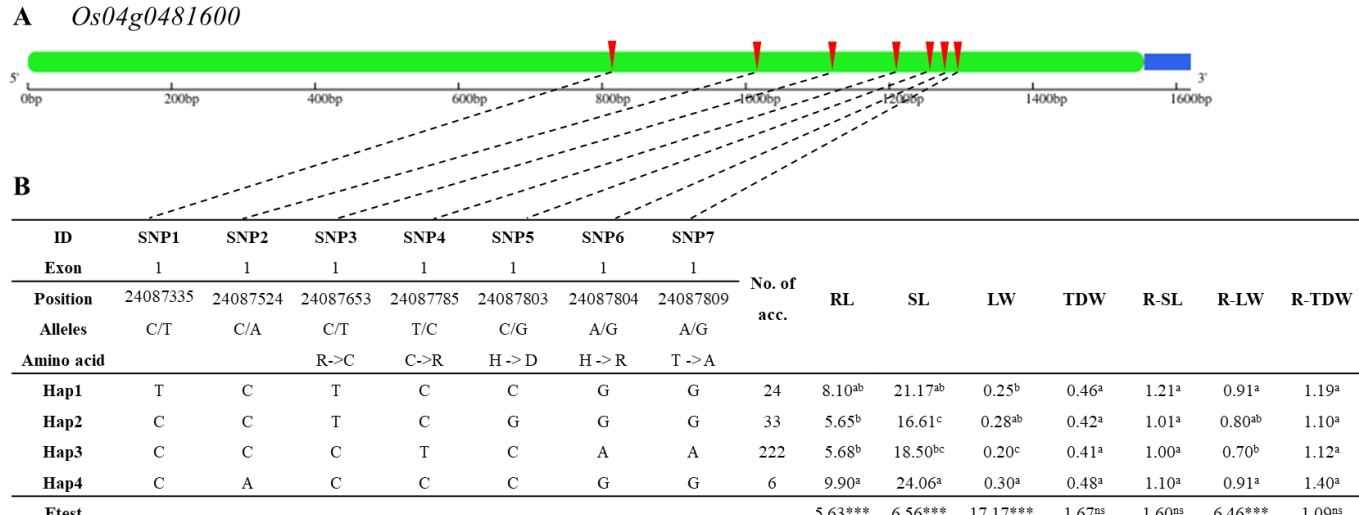

**A**　*Os04g0481600*

**B**

| ID | SNP1 | SNP2 | SNP3 | SNP4 | SNP5 | SNP6 | SNP7 | No. of acc. | RL | SL | LW | TDW | R-SL | R-LW | R-TDW |
|---|---|---|---|---|---|---|---|---|---|---|---|---|---|---|---|
| Exon | 1 | 1 | 1 | 1 | 1 | 1 | 1 | | | | | | | | |
| Position | 24087335 | 24087524 | 24087653 | 24087785 | 24087803 | 24087804 | 24087809 | | | | | | | | |
| Alleles | C/T | C/A | C/T | T/C | C/G | A/G | A/G | | | | | | | | |
| Amino acid | | | R->C | C->R | H->D | H->R | T->A | | | | | | | | |
| Hap1 | T | C | T | C | C | G | G | 24 | 8.10[ab] | 21.17[ab] | 0.25[b] | 0.46[a] | 1.21[a] | 0.91[a] | 1.19[a] |
| Hap2 | C | C | T | C | G | G | G | 33 | 5.65[b] | 16.61[c] | 0.28[ab] | 0.42[a] | 1.01[a] | 0.80[ab] | 1.10[a] |
| Hap3 | C | C | C | T | C | A | A | 222 | 5.68[b] | 18.50[bc] | 0.20[c] | 0.41[a] | 1.00[a] | 0.70[b] | 1.12[a] |
| Hap4 | C | A | C | C | C | G | G | 6 | 9.90[a] | 24.06[a] | 0.30[a] | 0.48[a] | 1.10[a] | 0.91[a] | 1.40[a] |
| Ftest | | | | | | | | | 5.63*** | 6.56*** | 17.17*** | 1.67[ns] | 1.60[ns] | 6.46*** | 1.09[ns] |

**C**

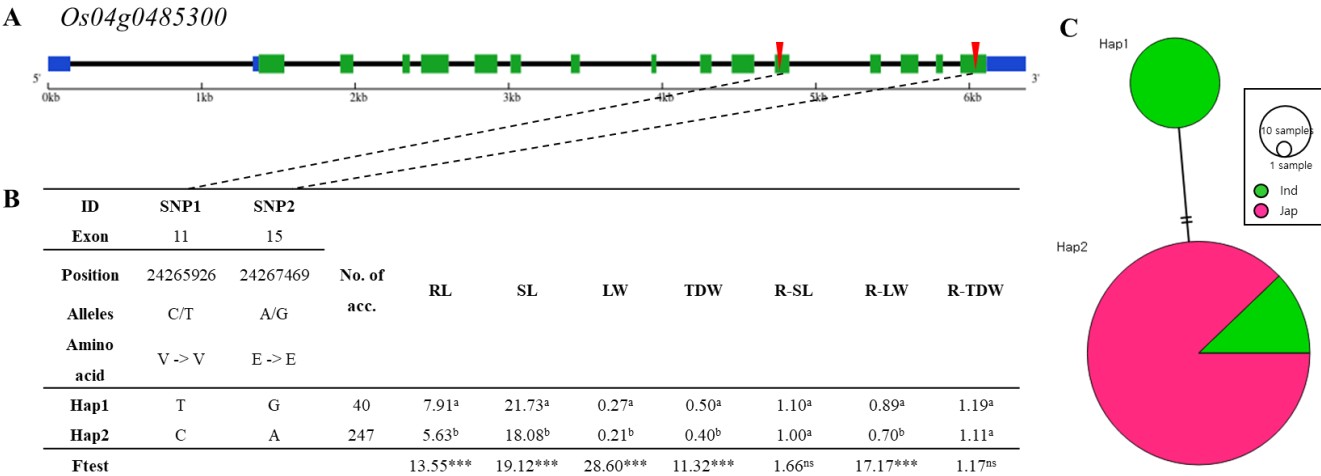

**Figure 6.** Haplotype analysis of *Os04g0481600*. (**A**) Gene structure and SNP positions on *Os04g0481600*. Green, blue, and red represent exon, untranslated region (UTR), and SNPs, respectively. (**B**) Significant haplotypes by ANOVA at *** $p < 0.001$. a, b and c indicate different levels, and ab indicates same level of Duncan's test. RL, SL, LW, TDW, R-SL, R-LW, and R-TDW refer to average value of root length, shoot length, leaf width, total dry weight, relative shoot length, relative leaf width, and relative total dry weight, respectively. (**C**) Haplotype variation analysis. Colors indicate rice subspecies, as indicated in legend. Circle size represents number of varieties in each Hap. Transverse lines show extent of variation between two Haps.

**A**　*Os04g0485300*

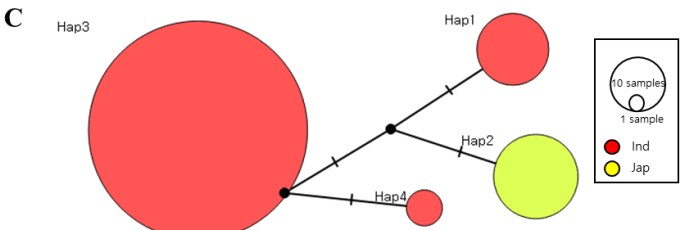

**C**

**B**

| ID | SNP1 | SNP2 | No. of acc. | RL | SL | LW | TDW | R-SL | R-LW | R-TDW |
|---|---|---|---|---|---|---|---|---|---|---|
| Exon | 11 | 15 | | | | | | | | |
| Position | 24265926 | 24267469 | | | | | | | | |
| Alleles | C/T | A/G | | | | | | | | |
| Amino acid | V -> V | E -> E | | | | | | | | |
| Hap1 | T | G | 40 | 7.91[a] | 21.73[a] | 0.27[a] | 0.50[a] | 1.10[a] | 0.89[a] | 1.19[a] |
| Hap2 | C | A | 247 | 5.63[b] | 18.08[b] | 0.21[b] | 0.40[b] | 1.00[a] | 0.70[b] | 1.11[a] |
| Ftest | | | | 13.55*** | 19.12*** | 28.60*** | 11.32*** | 1.66[ns] | 17.17*** | 1.17[ns] |

**Figure 7.** Haplotype analysis of *Os04g0485300*. (**A**) Gene structure and SNP positions on *Os04g0485300*. Green, blue, black, and red indicate exon, untranslated region (UTR), intron, and SNPs, respectively. (**B**) Significant haplotypes by ANOVA at *** $p < 0.001$. a, and b indicate different levels of Duncan's test. RL, SL, LW, TDW, R-SL, R-LW, and R-TDW refer to average value of root length, shoot length, leaf width, total dry weight, relative shoot length, relative leaf width, and relative total dry weight, respectively. (**C**) Haplotype variation analysis. Colors indicate rice subspecies, as indicated in legend. Circle size represents number of varieties in each Hap. Transverse lines show extent of variation between two Haps.

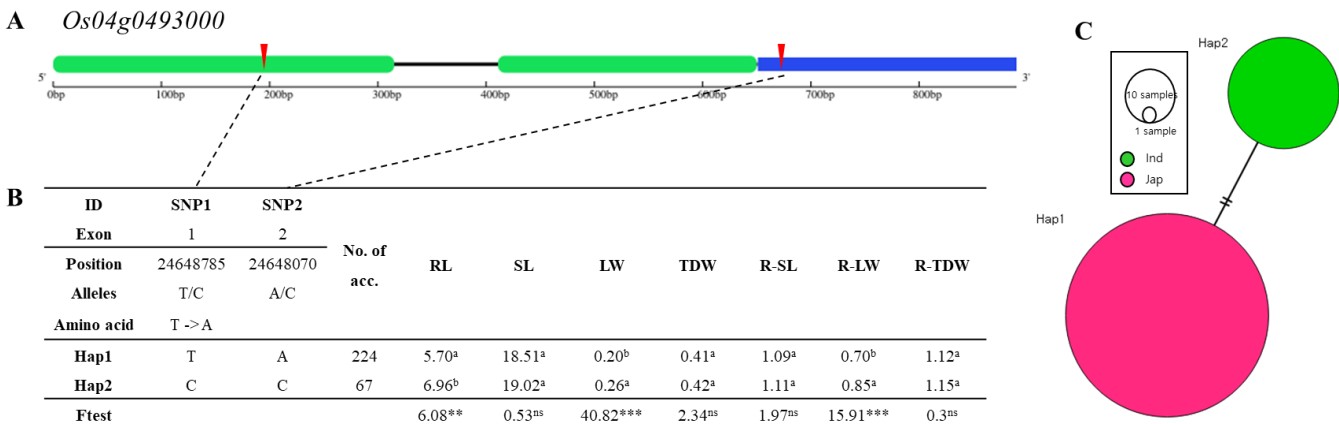

**A** *Os04g0493000*

**B**

| ID | SNP1 | SNP2 | No. of acc. | RL | SL | LW | TDW | R-SL | R-LW | R-TDW |
|---|---|---|---|---|---|---|---|---|---|---|
| Exon | 1 | 2 | | | | | | | | |
| Position | 24648785 | 24648070 | | | | | | | | |
| Alleles | T/C | A/C | | | | | | | | |
| Amino acid | T -> A | | | | | | | | | |
| Hap1 | T | A | 224 | 5.70$^a$ | 18.51$^a$ | 0.20$^b$ | 0.41$^a$ | 1.09$^a$ | 0.70$^b$ | 1.12$^a$ |
| Hap2 | C | C | 67 | 6.96$^b$ | 19.02$^a$ | 0.26$^a$ | 0.42$^a$ | 1.11$^a$ | 0.85$^a$ | 1.15$^a$ |
| Ftest | | | | 6.08** | 0.53$^{ns}$ | 40.82*** | 2.34$^{ns}$ | 1.97$^{ns}$ | 15.91*** | 0.3$^{ns}$ |

**Figure 8.** Haplotype analysis of *Os04g0493000*. (**A**) Gene structure and SNP positions on *Os04g0493000*. Green, blue, black, and red represent exon, untranslated region (UTR), intron, and SNPs, respectively. (**B**) Significant haplotypes by ANOVA at *** $p < 0.001$ and ** $p < 0.01$. a and b indicate different levels of Duncan's test. RL, SL, LW, TDW, R-SL, R-LW, and R-TDW refer to average value of root length, shoot length, leaf width, total dry weight, relative shoot length, relative leaf width, and relative total dry weight, respectively. (**C**) Haplotype variation analysis. Colors indicate rice subspecies, as indicated in legend. Circle size represents number of varieties in each Hap. Transverse lines show extent of variation between two Haps.

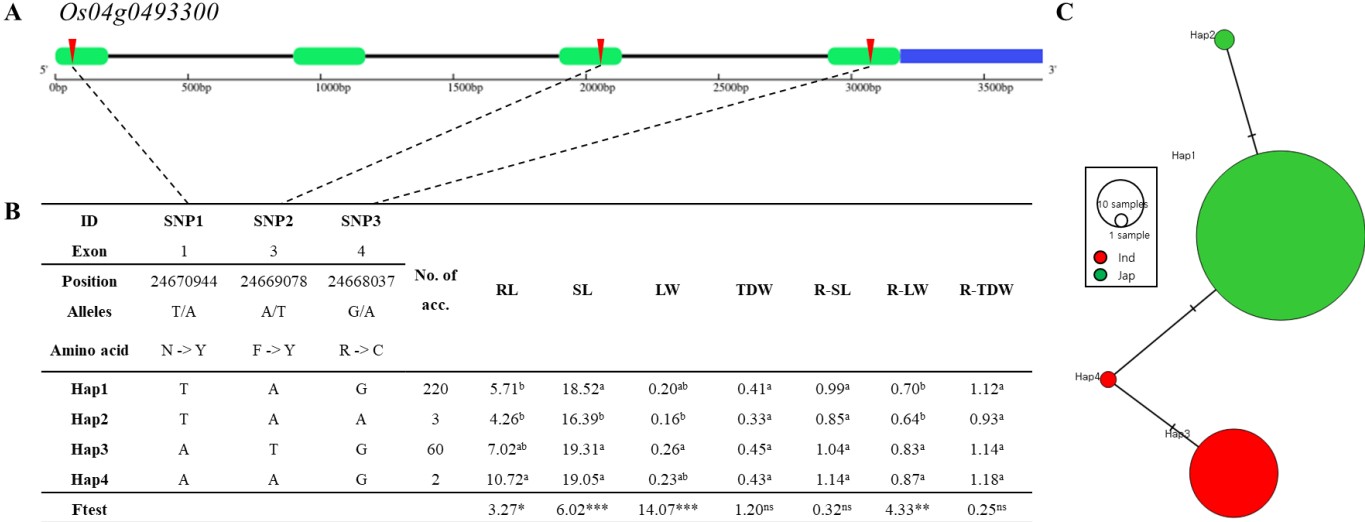

**A** *Os04g0493300*

**B**

| ID | SNP1 | SNP2 | SNP3 | No. of acc. | RL | SL | LW | TDW | R-SL | R-LW | R-TDW |
|---|---|---|---|---|---|---|---|---|---|---|---|---|
| Exon | 1 | 3 | 4 | | | | | | | | |
| Position | 24670944 | 24669078 | 24668037 | | | | | | | | |
| Alleles | T/A | A/T | G/A | | | | | | | | |
| Amino acid | N -> Y | F -> Y | R -> C | | | | | | | | |
| Hap1 | T | A | G | 220 | 5.71$^b$ | 18.52$^a$ | 0.20$^{ab}$ | 0.41$^a$ | 0.99$^a$ | 0.70$^b$ | 1.12$^a$ |
| Hap2 | T | A | A | 3 | 4.26$^b$ | 16.39$^b$ | 0.16$^b$ | 0.33$^a$ | 0.85$^a$ | 0.64$^b$ | 0.93$^a$ |
| Hap3 | A | T | G | 60 | 7.02$^{ab}$ | 19.31$^a$ | 0.26$^a$ | 0.45$^a$ | 1.04$^a$ | 0.83$^a$ | 1.14$^a$ |
| Hap4 | A | A | G | 2 | 10.72$^a$ | 19.05$^a$ | 0.23$^{ab}$ | 0.43$^a$ | 1.14$^a$ | 0.87$^a$ | 1.18$^a$ |
| Ftest | | | | | 3.27* | 6.02*** | 14.07*** | 1.20$^{ns}$ | 0.32$^{ns}$ | 4.33** | 0.25$^{ns}$ |

**Figure 9.** Haplotype analysis of *Os04g0493300*. (**A**) Gene structure and SNP positions on *Os04g0493300*. Green, blue, black, and red represent exon, untranslated region (UTR), intron, and SNPs, respectively. (**B**) Significant haplotypes by ANOVA at *** $p < 0.001$, ** $p < 0.01$, and * $p < 0.05$. a and b indicate different levels, and ab means same level of Duncan's test. RL, SL, LW, TDW, R-SL, R-LW, and R-TDW refer to average value of root length, shoot length, leaf width, total dry weight, relative shoot length, relative leaf width, and relative total dry weight, respectively. (**C**) Haplotype variation analysis. Colors indicate rice subspecies, as indicated in legend. Circle size represents number of varieties in each Hap. Transverse lines show extent of variation between two Haps.

*Os04g0493300* encodes glycine-rich protein and three SNPs in exon regions (Figure 9A). This gene contains three significant SNPs (T→A, Chr4_24670944, N→Y substitution; A→T, Chr4_24669078, F→Y substitution; G→A, Chr4_24668037, R→C substitution) that contain four haplotypes (Figure 9B). Significant differences among haplotypes varying in *Os04g049330* for root length, shoot length, leaf width, and relative leaf width were detected. The relative leaf width of Hap1 of *Os04g0493300* differed significantly from Hap2 and Hap4, and maximum root length variation was 6.46. The haplotype of Hap1 and Hap2 consist of *japonica* varieties, and the haplotype of Hap3 and Hap4 consist of *indica* varieties (Figure 9C).

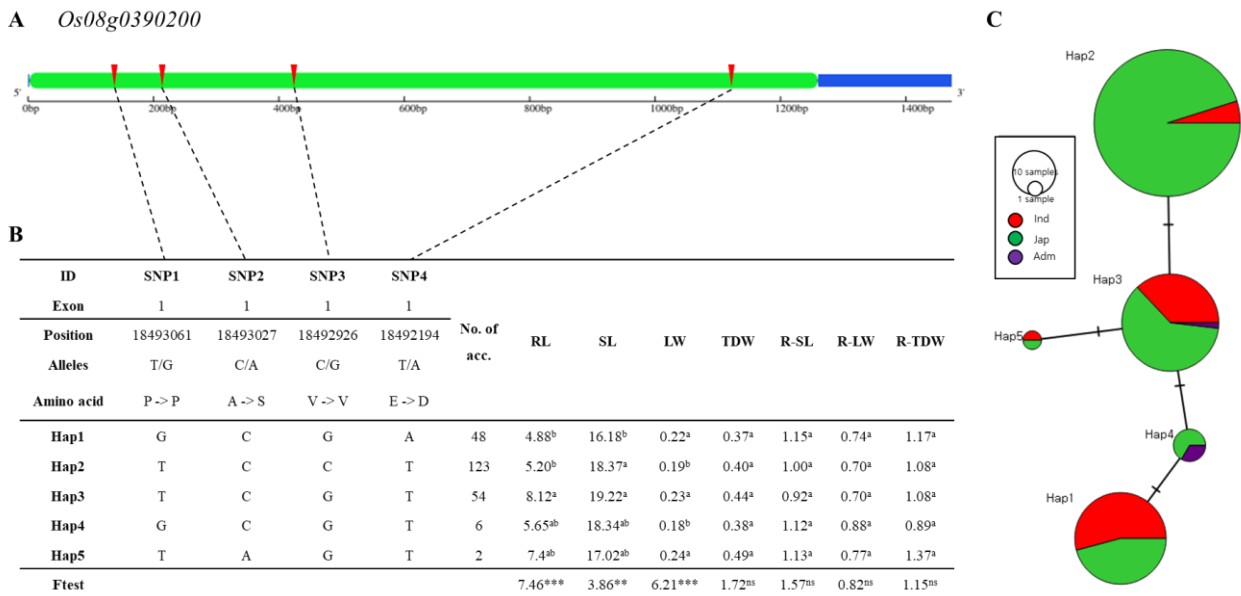

**Figure 10.** Haplotype analysis of *Os08g0390200*. (**A**) Gene structure and SNP positions on *Os08g0390200*. Green, blue, black, and red represent exon, untranslated region (UTR), intron, and SNPs, respectively. (**B**) Significant haplotypes by ANOVA at *** $p < 0.001$ and ** $p < 0.01$. a and b indicate different levels, and ab means same level of Duncan's test. RL, SL, LW, TDW, R-SL, R-LW, and R-TDW refer to average value of root length, shoot length, leaf width, total dry weight, relative shoot length, relative leaf width, and relative total dry weight respectively. (**C**) Haplotype variation analysis. Colors indicate rice subspecies, as indicated in legend. Circle size represents number of varieties in each Hap. Transverse lines show extent of variation between two Haps.

The gene *Os08g0390200* encodes 1-aminocyclopropane-1-carboxylate oxidase homolog 4 protein (Figure 10A). This gene includes only one exon, but four non-synonymous SNPs. Two of the four SNPs have changed alleles, with amino acid substitutions (C→A, Chr8_18493027, A→S substitution; T→A, Chr8_18492194, E→D substitution). The other two SNPs have changed alleles but not changed amino acids (T→G, Chr8_18493061, P→P; C→G, Chr8_18492926, V→V). *Os08g0390200* is composed of five haplotype groups (Figure 10B). Significant differences among haplotypes varying in *Os08g0390200* for root length, shoot length, and leaf width were detected. For example, the root length of Hap1 of *Os08g0390200* differed significantly from Hap3 and Hap5, and maximum root length variation was 3.24. The haplotype of Hap1, Hap2 and Hap5 consist of *japonica* and *indica* varieties. The haplotype of Hap4 consist of *japonica* and *admixture* varieties and the haplotype of Hap3 consists of *indica*, *japonica* and *admixture* (Figure 9C). The number of individuals assigned to Hap 2 is the largest, with 123 accessions (6 *indica*, 117 *japonica*). The smallest number of individuals is Hap 5, which consists of one *indica* and one *japonica* accession (Figure 10C).

## 4. Discussion

The world's population continues to increase, requiring more rice production. However, global salinization is also increasing and we face many limitations, including limited parental resources for conventional breeding and the complexity of salinity tolerance in rice [49]. To develop salt-tolerant varieties, molecular marker techniques and biotechnology are being used in combination with conventional breeding methods [50–52].

We compared the physical/genetic regions of the QTLs identified in this study with previously reported QTLs. The results revealed that the region of *qSIS4* QTL was overlapped with that of AQCI0118 QTL which is associated with phosphorus sensitivity. The region of *qSIS8* QTL was overlapped with that of AQDP005 QTL which is associated with iron sensitivity. Unexpectedly, there was no previously reported QTLs for salt tolerance in the regions of currently detected QTLs. The result of the overlapped regions

among the two previously QTLs for performance of root and currently detected QTLs for salt tolerance suggests that there may be some overlapped pathway between salt tolerance and root performance. To further narrow down the candidate genes in the QTL regions, we utilized a database of gene expression studies [41] and GWAS for salt tolerance [23]. From the gene expression database, we detected two candidate genes (*Os04g0481600* and *Os08g0390200*) which were specifically upregulated in salt tolerant varieties under the salt treatment. The haplotype analysis for these two genes, performed with the database of a previously reported GWAS for salt tolerance, showed significant differences for the phenotypic performance under salt conditions of rice cultivars carrying different haplotypes of *Os04g0481600* and *Os08g0390200*, therefore providing additional indirect evidence for their possible role in salt tolerance. The candidate *Os04g0481600* gene encodes WD domain, G-beta repeat domain containing protein. Five salt responsive WD40 proteins (SRWDs) were reported, which were specifically expressed in leaf or root under salt treatment. For example, expression of SRWD1 is regulated with different responsive patterns in leaves and roots of tolerant cultivar Jiucaiqing and sensitive cultivar IR26 under salt stress [53]. In wheat, a number of TaWD40 genes respond to abiotic stress such as cold, heat, or drought. In addition, specific expression of TaWD40 genes against abiotic stress such as powdery mildew or rust pathogen infection was reported [54]. *O8g0390200* gene encodes ACO, which is involved in the final step of ethylene production in plant tissues [55]. Ethylene is a gaseous plant hormone that regulates all physiological processes during the plant's life cycle. Many ACO genes have been isolated from different plant species, and the expression of those genes have been found to vary depending on the tissue, developmental stage, and environmental conditions [56]. A wheat TaACO1 negatively regulated salinity stress in *Arabidopsis thaliana* [57]. Additionally, we selected three more candidate genes (*Os04g0485300*, *Os04g0493300*, and *Os04g0493000*) among the 44 specifically expressed genes against the salt treatment, *Os04g0485300* encodes a glucose-6-phosphate 1-dehydrogenase (G6PDH), *Os04g0493300* encodes glycine-rich proteins (GRPs), and *Os08g0390200* encodes 1-aminocyclopropane-1-carboxylate oxidase homolog 4 protein (ACO). Even though the expressions were downregulated in the tolerant variety under the salt treatment, we selected these three genes as candidate genes based on the relation between encoded proteins and salt tolerance previously reported in other plants. Interestingly, the haplotype analysis for these three genes with the database of previously reported GWAS for salt tolerance also showed significant difference among the phenotypic performance of haplotypes of *Os04g0485300*, *Os04g0493300*, and *O8g0390200*. Even though, no direct evidence that the down regulation of *Os04g0485300*, *Os04g0493300*, and *Os08g0390200* is associated with salt tolerance has not been reported in rice, in other plant, the encoded protein from these genes were reported to be related in various stress including salt stress. Moreover, the negative effect of GRPs on salt stress conditions in *Camelina sativa* [58] and negative regulation of TaACO1 on salt stress in Arabidopsis thaliana were reported, suggesting the possible role of downregulation of *Os04g0493300* and *Os08g0390200* in salt tolerance in rice. *Os04g0485300* encodes G6PDH, which was previously reported for its possible role in salt tolerance in soybean [45,46]. Zhao et al. [45] examined the physiological and transcriptional responses of GmG6PDH to different stresses in soybean, including salt, alkali, and osmotic stress. Significant induction was observed in all treatments, especially under salt stress. Transgenic soybean, including GmG6PDH2 overexpressing hairy roots (GmG6PDh2-OHR), showed significantly improved resistance to salinity stress at the seedling stage by increasing root fresh weight and root length. The roles of plant G6PDHs in the response to various abiotic stresses were reported, such as salinity tolerance in reed [46], cold tolerance in tobacco [59] and strawberry [60], drought tolerance in tomato [61], and heat tolerance in *Przewalskia tangutica* and tobacco [62]. Moreover, it was reported that G6PDH is involved as a regulator in maintaining cell redox balance in rice suspension cultures under salt stress [63]. *Os04g0493300* encodes GRPs. The protein encoded by GRP-1 consists of 384 amino acids, of which 67% are glycine [64]. Based on the arrangement of glycine repeats and the presence of conserved motifs, GRPs are divided

into five classes [65]. In Arabidopsis, GRP2, GRP4, and GRP7 genes are upregulated to enhance freezing tolerance compared with wild-type and grp-2 knockout mutants [66]. Kim et al. [67] reported that GRP2 enhanced seed germination under cold stress (11 °C) and influenced Arabidopsis growth in an ABA-dependent manner. Overexpression of AtGRP1 improved stress response under high salt conditions [68]. Overexpression of *Medicago sativa* GRPs in transgenic Arabidopsis showed retarded seed germination and seedling growth [69]. The candidate *Os04g0493000* encodes B-box (BBX) zinc finger family protein. The specific responses of BBX under abiotic stress in various crops were reported [70–74]. Genome-wide identification analysis of BBX genes in maize, rice, sorghum, millet, and stiff brome was conducted, and expression under abiotic (cold, drought, salt), hormonal (GA, ABA, SA, MeJA), and metal (Cr, Cd, Ni, Fe) stress in rice was significantly affected. The transcript level of most rice BBX genes was high at the heading stage, followed by the booting and seedling stages [72]. Overexpression of MdBBX10 in Arabidopsis induced enhanced tolerance to salt and drought stress, with a high germination ratio and long root, and improved the plant's ability to scavenge reactive oxygen species and conduct ABA signaling [48]. Moreover, the BBX family protein in chrysanthemum, CmBBX19, suppresses the expression of a set of stress and ABA responsive genes, such as CmRAB18 and CmRD29B, in normal conditions, but its expression is downregulated to release those genes to increase tolerance in drought stress conditions [74].

## 5. Conclusions

Through a genome-wide association study, we detected three QTLs for salt tolerance in rice seedlings. With the aid of bioinformatics analysis with a gene expression database and a previously reported salt stress experiment, we determined five candidate genes (*Os04g0481600*, *Os04g0485300*, *Os04g0493000*, *Os04g0493300* and *Os08g0390200*). Further studies evaluating the molecular role of these candidate genes will be carried out in near the future.

**Supplementary Materials:** The following are available online at https://www.mdpi.com/article/10.3390/agriculture11111174/s1, Table S1: Korean landrace accessions used in this study. Table S2: Candidate genes within 600 kb of significant SNP regions. Table S3: RNA-seq data of genes in candidate region.

**Author Contributions:** Conceptualization, S.-W.K. and J.L.; methodology, Y.L. and J.L.; resources, Y.L., Y.-J.P. and J.L.; data collection, F.-Y.C., N.-E.K. and J.-H.L.; validation, S.M.L., S.-G.J. and A.-R.L.; data analysis, S.M.L., J.S. and H.Z.; writing—original draft preparation, S.M.L.; writing—reviewing and editing, S.-W.K. and J.L.; supervision, project administration, and funding acquisition, S.-W.K. and J.L. All authors have read and agreed to the published version of the manuscript.

**Funding:** This research was funded by Rural Development Administration, Republic of Korea (Grant number PJ01480501 & PJ01579403).

**Institutional Review Board Statement:** Not applicable.

**Informed Consent Statement:** Not applicable.

**Data Availability Statement:** Not applicable.

**Conflicts of Interest:** The authors declare no conflict of interest.

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
