# Peer review of "Genome-Wide Association Study for Detecting Salt-Tolerance Loci and Candidate Genes in Rice"

_agriculture, doi:10.3390/agriculture11111174_

Round 1

Reviewer 1 Report

Lar et al. conducted an experiment to study the tolerance to salinity conditions during the seedling stage of Korean rice cultivars. The achievement of this work is to identify three QTLs significantly associated to variation in salt tolerance of rice seedlings. Within these QTLs, candidate genes and haplotypes are investigated by using expression, genotypic and phenotypic data sets previously reported in data bases. The main concern I have is that authors did not validate the expression data with their own plant material. I mean, they could have performed qPCR experiments for the candidate genes using the two Korean varieties with the most contrasting phenotypes under salt conditions to confirm the putative relationship between the observed phenotypic variation and differences in the expression pattern of these genes.  

Some suggestions:

Line 16: to understand the genetic basis of salt tolerance at the seedling stage in Korean rice.

Line 34: add accession date.

Line 40: pH higher than 8.5

Line 43: link doesn’t work

Line 46: at both the seedling and reproductive stages.

Line 52: major symptoms of salinity at the reproductive stage?

Line 149: performed using R?

Line 172: three data sources were used for candidate gene expression patterns? Please give more details. In Line 249, authors refer only to one data set.

Line 180: data sets used for haplotype analysis should be described here.

Line 200: which three groups? You have not described previously any groups among accessions.

Line 206: could you describe if there is any trait or origin of the accessions that can explain the population structure determined?

Line 208: principal component analysis.

Line 236: QTL instead of QTLs.

Line 237-242: Could you rewrite these sentences in one?

Line 250: significantly differentially expressed.

Line 252: … 18 genes were upregulated under salt stress conditions. In the susceptible IR29 variety growing under salinity stress conditions, 11 genes were upregulated, while the expression of these genes was not increased in the tolerant variety Pokkali.

Line 255: which seven genes were overexpressed only in the tolerant variety? it could be interesting to describe them and to justify better why only two were investigated. It is surprising that authors have chosen these two genes because they are upregulated under salt conditions in the tolerant variety and did not study other five genes with the same opposite expression pattern, while studied three other genes (Os04g0485300, Os04g0493000 and Os04g0493300) that are downregulated in the tolerant variety.

Line 267: it could be useful to mark (i. e. *) the selected genes in the Figure 5

Line 280: I think that several expressions used along the results section are not correct. Maybe better here: “We detected significant differences among haplotypes varying in the five candidate genes for at least three out of the six phenotypic traits (root length, shoot length, …) studied”?

Lines 289 and 305: explain the significant differences between haplotypes (figures 6B and 7B).

Lines 289 and 319. How two haplotypes are highly significantly different? Significant differences are detected for a trait, not for nucleotides.  And secondly: do you mean that the haplotype 1 is found only in varieties from the japonica group and the haplotype 2 is present only in indica varieties? I think that some explanation about the phenotypic differences observed (Figure 8B) should be included.

Line 332. When growing in salt stress conditions, root length of varieties carrying haplotype 2 are significantly different from root length of varieties carrying haplotype 4. Or varieties carrying the haplotype 4 for this gene showed root length values significantly higher than those with the haplotype 2 when cultivated in 200 mM salt. Why authors didn’t mention significant differences observed for other traits? If they are not important, why were included in Figures 6B, 7B, 8B, and 9B?

Line 348. Again, authors did not describe phenotypic variation observed among phenotypes (Figure 10B). This is quite surprising, since this is the only evidence they present for the possible role of the candidate genes in rice salt tolerance.

Line 364: are being used in combination…

Line 370: where were reported? Citation needed?

Line 376: I think that the discussion section should be rewritten, since it is merely a sum of references. Authors discuss results from others but not those from their own work. There is nothing about the GWAS performed. I think they could discuss the phenotypic variation observed among haplotypes, but this information is not described in the previous results section.

Line 427: Based on the previously reported function of these genes in various plants, their possible role in salt tolerance in rice seedlings is supported. Yes, and this was true before this work, therefore this cannot be a conclusion of this work.

Line 455: reference 6 is incomplete.  In book: Agricultural Drainage (pp.55-108) Chapter: Crop yields as affected by salinity Publisher: American Society of Agronomy Editors: R. W. Skaggs and J. van Schilfgaarde

Line 449, 458, 485: Oryza sativa should be in italics.

Line 450: reference 4 is incomplete and doesn’t follow journal’s format.

Line 494: japonica should be in italics.

Line 503: Hordeum spontanum should be in italics.

Line 519: reference 32 is incomplete.

Line 563: reference 52 doesn’t follow journal format.

Line 566: Oryza sativa should be in two words and in italics.

Line 568: Triticum aestivum should be in italics.

Line 570: Populus suaveolens should be in italics

Line 576: Solanum lycopersicum should be in italics.

Line 579: Przewalskia tangutica and Nicotiana tabacum in italics.

Line 600: Escherichia coli in italics.

Lines 603, 621: Arabidopsis thaliana in italics.

Line 609: Camelina sativa in italics.

Author Response

Thank you for your sincere review and valuable comments for improving this manuscript. We revised according your comments.

Line 16: to understand the genetic basis of salt tolerance at the seedling stage in Korean rice.

 -We revised it

Line 34: add accession date.

-We are not sure how to cite the data of web page, we will revise it after consulting the editing office in the final version.

Line 40: pH higher than 8.5

-We revised it

Line 43: link doesn’t work

-We revised it

Line 46: at both the seedling and reproductive stages.

-We revised it

Line 52: major symptoms of salinity at the reproductive stage?

-We revised it

Line 149: performed using R?

-We revised it

Line 172: three data sources were used for candidate gene expression patterns? Please give more details. In Line 249, authors refer only to one data set.

-We revised it to make it clear       

lLine 180: data sets used for haplotype analysis should be described here.

 -We revised it

Line 200: which three groups? You have not described previously any groups among accessions.

-We just described that our 191 landraces were clearly separated into 3 groups (cluster)s. we revise it as cluster not group

Line 206: could you describe if there is any trait or origin of the accessions that can explain the population structure determined?

-As we mentioned the previously, our materials are Korean landrace obtained from other institution, we don’ have other specific information on the taxonomy. 

Line 208: principal component analysis.

-We revised it

Line 236: QTL instead of QTLs.

-We revised it

Line 237-242: Could you rewrite these sentences in one?

-It will be Ok to make it one sentence however, it will be OK to keep this as it is. We would like to keep those as separate sentences

Line 250: significantly differentially expressed.

-We revised it

Line 252: … 18 genes were upregulated under salt stress conditions. In the susceptible IR29 variety growing under salinity stress conditions, 11 genes were upregulated, while the expression of these genes was not increased in the tolerant variety Pokkali.

-We made mistake in writing the manuscript.

We corrected the mistake as bellow

“Among the 44 differentially expressed genes, 19 genes were increased under salt stress conditions. In the susceptible IR29 variety under salt-treated conditions, 12 genes were increased; these genes were not increased in the tolerant Pokkali variety. Five genes were increased in both susceptible and tolerant varieties, and two genes were increased only in the tolerant variety”.

Line 255: which seven genes were overexpressed only in the tolerant variety? it could be interesting to describe them and to justify better why only two were investigated. It is surprising that authors have chosen these two genes because they are upregulated under salt conditions in the tolerant variety and did not study other five genes with the same opposite expression pattern, while studied three other genes (Os04g0485300, Os04g0493000 and Os04g0493300) that are downregulated in the tolerant variety.

-As we mentioned previously, we made mistake to write the manuscript. Only two gene was increased in only POK, We designate these two genes as candidate genes and we conducted the haplotype analysis. In addition, we proposed three more candidate genes (Os04g0485300, Os04g0493000, and Os04g0493300). Based on the previously reported function, The encoded proteins from these genes were reported to be associated with salt tolerance in other plants. Even though, the expression of these genes in POK(tolerant variety) was decreased, there is still the possibility for the association between salt tolerance and gene downregulation. Thus we proposed these three genes as candidate genes.  We would like to share the information of the candidate gene as much as we can, thus we think it would be better for readers to present these five candidate genes in this manuscript and they can evaluate the function of these candidate genes if they want to. We revised these in the discussion part

Line 267: it could be useful to mark (i. e. *) the selected genes in Figure 5

-We marked selected genes.

Line 280: I think that several expressions used along the results section are not correct. Maybe better here: “We detected significant differences among haplotypes varying in the five candidate genes for at least three out of the six phenotypic traits (root length, shoot length, …) studied”?

-We revised it as the reviewer’s suggestion.

Lines 289 and 305: explain the significant differences between haplotypes (Figures 6B and 7B).

 -We revised it

Lines 289 and 319. How two haplotypes are highly significantly different? Significant differences are detected for a trait, not for nucleotides. And secondly: do you mean that haplotype 1 is found only in varieties from the japonica group and haplotype 2 is present only in indica varieties? I think that some explanation about the phenotypic differences observed (Figure 8B) should be included.

 -We mentioned the difference in each phenotype. 

Line 332. When growing in salt stress conditions, the root length of varieties carrying haplotype 2 is significantly different from the root length of varieties carrying haplotype 4. Or varieties carrying the haplotype 4 for this gene showed root length values significantly higher than those with the haplotype 2 when cultivated in 200 mM salt. Why authors didn’t mention significant differences observed for other traits? If they are not important, why were included in Figures 6B, 7B, 8B, and 9B?

-We revised these as the reviewer’s suggestion.

Line 348. Again, authors did not describe phenotypic variation observed among phenotypes (Figure 10B). This is quite surprising, since this is the only evidence they present for the possible role of the candidate genes in rice salt tolerance.

-We revised it

Line 364: are being used in combination…

-We revised it

Line 370: where were reported? Citation needed?

-We mentioned in the introduction We deleted that sentence instead of adding a citation.

Line 376: I think that the discussion section should be rewritten, since it is merely a sum of references. Authors discuss results from others but not those from their own work. There is nothing about the GWAS performed. I think they could discuss the phenotypic variation observed among haplotypes, but this information is not described in the previous results section.

-We reorganize and revised the discussion part. We tried to support the possible role of the candidate gene with the previously reported function. Anyway as the reviewer’s suggestion we rewrote the discussion part.

Line 427: Based on the previously reported function of these genes in various plants, their possible role in salt tolerance in rice seedlings is supported. Yes, and this was true before this work, therefore this cannot be a conclusion of this work.

-We deleted it

Line 455: reference 6 is incomplete.  In book: Agricultural Drainage (pp.55-108) Chapter: Crop yields as affected by salinity Publisher: American Society of Agronomy Editors: R. W. Skaggs and J. van Schilfgaarde

-We edited the reference

Line 449, 458, 485: Oryza sativa should be in italics.

-We edited the reference

Line 450: reference 4 is incomplete and doesn’t follow journal’s format.

-We edited the reference

Line 494: japonica should be in italics.

-We edited the reference

Line 503: Hordeum spontanum should be in italics.

-We edited the reference

Line 519: reference 32 is incomplete.

-We edited the reference

Line 563: reference 52 doesn’t follow journal format.

-We edited the reference

Line 566: Oryza sativa should be in two words and in italics.

-We edited the reference

Line 568: Triticum aestivum should be in italics.

-We edited the reference

Line 570: Populus suaveolens should be in italics

-We edited the reference

Line 576: Solanum lycopersicum should be in italics.

-We edited the reference

Line 579: Przewalskia tangutica and Nicotiana tabacum in italics.

-We edited the reference

Line 600: Escherichia coli in italics.

-We edited the reference

Lines 603, 621: Arabidopsis thaliana in italics.

-We edited the reference

Line 609: Camelina sativa in italics.

-We edited the reference

Reviewer 2 Report

In the manuscript titled "Genome-Wide Association Study for Detecting Salt-Tolerance 2 Loci and Candidate Genes in Rice", the authors assessed the salt tolerance for 191 Korean landrace rice accessions and conducted a GWAS analysis with SNPs derived from a gene-Chip based Method. They detected three QTLs and identified five candidate genes that may be associated with salt tolerance for their samples. Overall, the manuscript is good written and the result is clearly presented. I don’t have any question about their analyses and results.

Author Response

Thank you for your review.

Author Response

1. The quality of Figure1 picture is too poor to meet the basic requirements of the paper. Please re-tailor it again.

-We refined Figure 1.

2 Figure 4 needs to mark the threshold line in the diagram.

- we did it.

3. Some spelling mistakes in the line-288,Chr_24087804 and Chr_24087809.

- We did it

4. Based on the expression analysis, the RNA level of Os04g0485300, Os04g0493000 and Os04g0493300 were decreased in POK when treated with salt, it is evident that they are nothing to do with salt tolerance of POK. So, I wonder why they were considered as the candidate genes?

- We proposed three more candidate genes (Os04g0485300, Os04g0493000, and Os04g0493300). Based on the previously reported function, The encoded proteins from these genes were reported to be associated with salt tolerance in other plants. Even though, the expression of these genes in POK(tolerant variety) was decreased, there is still the possibility for the association between salt tolerance and gene downregulation. Thus we proposed these three genes as candidate genes. in the discussion part, the previous report of negative regulation of Os04g0493000, and Os04g0493300 were discussed.   We would like to share the information of the candidate gene as much as we can, thus we think it would be better for readers to present these five candidate genes in this manuscript and they can evaluate the function of these candidate genes if they want to. We revised these in the discussion part

5. What is the relationship between haplotype analysis and the screening of salt-resistant genes in the manuscript? How about their effects on the conclusion of the article?

- The haplotype analysis was conducted with the previously reported dataset, thus the population is different. Interestingly, the results were in accordance with them. Since we only focused on the 5 candidate genes, it is difficult to mention of the relationship between haplotype analysis and the screening of salt-resistant genes. we would like to just present these results without discussing the relationship.  

6. The format of the literature should be reedited to keep consistent with requirements of the Journal.

- We revised it

7. In the section of Discussion, the authors just described the potential function of each genes based on the annotated homologous genes in other plants, which didn’t deepen the theme of this article.

-We revised most of the discussion part

8. The manuscript should be improved in the spelling and grammar of English, and make clear the promising results.

- we checked grammatical errors again and this manuscript was English edited by the English editing service company suggested by MDPI

Round 2

Reviewer 1 Report

In the manuscript “Genome-Wide Association Study for Detecting Salt-Tolerance Loci and Candidate Genes in Rice”, authors have performed a GWAS using Korean rice landraces to detect QTLs associated to salt tolerance. For this study, the phenotype of plants under salt conditions was described by a visual damage score. Candidate genes within the 3 tagged regions were analysed by using RNASeq and haplotype and phenotypic (6 growth-related traits) data sets obtained from previously published works.

From the RNASeq data, authors found in the tagged QTLs 44 genes reported as differentially expressed under salt conditions in two rice varieties, a salt-tolerant cultivar and a salt-sensitive one. They conclude that five of them are candidate genes for salt tolerance because: (a) 2 genes (Os04g0481600 and Os08g0390200) are up-regulated under salt-conditions in the tolerant cultivar while remained unchanged in the sensitive variety and, (b) 3 genes (Os04g0485300, Os04g0493000 and Os04g0493300) are down-regulated under stress in the two varieties, but are still proposed as candidate genes based on their previously reported function. Therefore, one should conclude that there are no other genes which had been described as involved in stress responses among the remaining 39 genes that are differentially expressed too, because this is the only evidence to support that these 3 genes could be involved in salt tolerance and therefore deserve further study.

But anyway, as I mentioned in my first review, these RNASeq expression profiles are not from varieties used to detect the QTLs. Why authors did not study and validate the patterns of expression of these genes in ja046-Jangsamdo and ja110-Jwiiparibyeo, the Korean cultivars they described as tolerant and susceptible, respectively?

From the haplotype and phenotype data sets (different again from those of the GWAS), authors conclude that different haplotypes of the five proposed genes are related to growth variation under salt conditions. Phenotypic traits used are not described in the Material and Methods section, as I also mentioned in my first review. Root length, shoot length, leaf width, and total dry weight data were determined by Yu et al. (2017) under salt conditions, but can be biased by among cultivars intrinsic differences. Authors should have discussed here the variation on relative shoot length, relative leaf width, and relative total dry weight, which describe plant growth decrease observed under salt conditions, and are the estimates used by Yu et al. to study the degree of salt tolerance.

After the revision performed by the authors, I still miss in the manuscript some discussion about the GWAS results.

Some other points:

Line 151: please correct the sentence “were performed using R package (version 4.03, http://r-project.org) [36], and for structural analysis we used ADMIXTURE software with bed format file.”

Line 271: add the explanation for the * in the figure legend.

Lines 385-: “We compared the physical/genetic positions of the QTLs identified in this study with previously reported QTLs revealed that the qSIS4 QTL overlapped with the AQCI0118 QTL, which is associated with Phosphorus sensitivity and qSIS8 QTL overlapped with the AQDP005 QTL, which is associated with Iron sensitivity. Unexpectedly, there was no QTLs for salt tolerance was overlapped with the currently detected QTLs in this study. The two the overlapped QTLs whose associated traits are highly related with performance of root phenotype suggests that there may be some overlapped pathway between salt tolerance and root performance.” I am not a native English-speaking person, but I think the manuscript needs a general revision. I copy these sentences as an example, since they are grammarly incorrect. I found many others that are difficult to understand, and that I think someone could re-write in an easier way for readers.

Line 391: we utilized a database

Line 394: “The haplotype analysis for these two genes with the database of previously reported GWAS for salt tolerance showed significant difference among the phenotypic performance of haplotypes of Os04g0481600 and Os04g0493000, providing additional indirect evidence for the possible role in salt tolerance”. Did you mean that? “The haplotype analysis for these two genes, performed with the database of a previously reported GWAS for salt tolerance, showed significant differences for the phenotypic performance under salt conditions of rice cultivars carrying different haplotypes of Os04g0481600 and Os04g0493000, therefore providing additional indirect evidence for their possible role in salt tolerance.”

Line 412: Please correct “long root length”

Line 418: The sentence “among the 44 specifically expressed genes in salt tolerant varieties under the salt treatment” it’s not true. These 44 genes were differentially expressed (up- or down-regulated) under the salt treatment as compared to control conditions in two varieties, a tolerant cultivar and a sensitive one. There are no data about the expression of these genes in a panel of rice varieties. Furthermore, in Line 422, authors wrote “Even though the expressions were downregulated in the tolerant variety under the salt treatment”.

Line 419: Os04g0485300 encodes a glucose-6-phosphate 1-dehydrogenase (G6PDH)…

Line 445: “maintaining cell redox balance in rice suspension cultures”

Line 446: I suggest to delete these redundant sentences: “Plant GRPs are characterized by a high content of glycine (20-70%). GRP was first discovered in petunia gene.”

Lines 431 and 455 repeat the same information.

Lines 432 and 463 repeat the same information.

Line 466: “Through a genome-wide association study, we detected three QTLs for salt tolerance in rice seedlings. Bioinformatic analyses performed with …

Author Response

Thank you for your sincere review to improve the manuscript. We carefully revised the manuscript according to the reviewer's comments.  

In the manuscript “Genome-Wide Association Study for Detecting Salt-Tolerance Loci and Candidate Genes in Rice”, authors have performed a GWAS using Korean rice landraces to detect QTLs associated to salt tolerance. For this study, the phenotype of plants under salt conditions was described by a visual damage score. Candidate genes within the 3 tagged regions were analysed by using RNASeq and haplotype and phenotypic (6 growth-related traits) data sets obtained from previously published works.

From the RNASeq data, authors found in the tagged QTLs 44 genes reported as differentially expressed under salt conditions in two rice varieties, a salt-tolerant cultivar and a salt-sensitive one. They conclude that five of them are candidate genes for salt tolerance because: (a) 2 genes (Os04g0481600 and Os08g0390200) are up-regulated under salt-conditions in the tolerant cultivar while remained unchanged in the sensitive variety and, (b) 3 genes (Os04g0485300, Os04g0493000 and Os04g0493300) are down-regulated under stress in the two varieties, but are still proposed as candidate genes based on their previously reported function. Therefore, one should conclude that there are no other genes which had been described as involved in stress responses among the remaining 39 genes that are differentially expressed too, because this is the only evidence to support that these 3 genes could be involved in salt tolerance and therefore deserve further study.

But anyway, as I mentioned in my first review, these RNASeq expression profiles are not from varieties used to detect the QTLs. Why authors did not study and validate the patterns of expression of these genes in ja046-Jangsamdo and ja110-Jwiiparibyeo, the Korean cultivars they described as tolerant and susceptible, respectively?

-> We agreed with the reviewer’s suggestion. It will be better if we conduct RNASeq experiment with ja046-Jangsamdo and ja110-Jwiiparibyeo. We are preparing this experiment, but the results will come later, and we will report these results in the next manuscript. Our main goal of the expression analysis with ja046-Jangsamdo and ja110-Jwiiparibyeo will be to detect additional candidates (besides the candidate genes detected in this experiment). Thus the expression analysis will be reported in the next manuscript. We hope to report these results in the Agriculture journal again. 

Here, this is another reason why we did not report the of RNASeq experiment with ja046-Jangsamdo and ja110-Jwiiparibyeo. We think that expression profiling also can provide indirect evidence. We know that our candidate genes are supported by indirect evidence.

We think that providing additional indirect evidence cannot be the direct evidence for the role of the candidate gene in salt tolerance. Thus, we prefer to conduct to gene-editing experiment with candidate genes detected in this study in the further experiment.

From the haplotype and phenotype data sets (different again from those of the GWAS), authors conclude that different haplotypes of the five proposed genes are related to growth variation under salt conditions. Phenotypic traits used are not described in the Material and Methods section, as I also mentioned in my first review. Root length, shoot length, leaf width, and total dry weight data were determined by Yu et al. (2017) under salt conditions, but can be biased by among cultivars intrinsic differences. Authors should have discussed here the variation on relative shoot length, relative leaf width, and relative total dry weight, which describe plant growth decrease observed under salt conditions, and are the estimates used by Yu et al. to study the degree of salt tolerance.

-> We added the method for phenotyping by Yu et al. in section 2.7. Haplotype Analysis. 

 Yes. As the reviewer’s comment, the traits of relative shoot length, relative leaf width, and relative total dry weight are more reasonable in salt tolerance. However, we will not discuss this in the discussion part because the data were not created by ourselves, and we don’t want to criticize the data created by other researchers. Instead of discussing the usefulness among the traits, we briefly mentioned the difference of the R-LW among the Hap groups in four candidate genes (line282~285). In the original manuscript, we mentioned the trait of the root as an example. In the revised manuscript, we mentioned the trait of R-LW as an example. ( line 249, line 311, line 327). Since Os04g0493000 did not show a difference among the Hap group, we mentioned this in line 285.

After the revision performed by the authors, I still miss in the manuscript some discussion about the GWAS results.

-> Now GWAS is a common tool for rice breeders. The GWAS method we used in this study is very straightforward, thus, we think it is not necessary to discuss the GWAS results related to the method of analysis. In the discussion part results, we discussed our results (QTLs) and previously reported QTLs in the region of the detected QTLs.

Some other points:

Line 151: please correct the sentence “were performed using R package (version 4.03, http://r-project.org) [36], and for structural analysis we used ADMIXTURE software with bed format file.”

-> We corrected that

Line 271: add the explanation for the * in the figure legend.

-> We did it

Lines 385-: “We compared the physical/genetic positions of the QTLs identified in this study with previously reported QTLs revealed that the qSIS4 QTL overlapped with the AQCI0118 QTL, which is associated with Phosphorus sensitivity and qSIS8 QTL overlapped with the AQDP005 QTL, which is associated with Iron sensitivity. Unexpectedly, there was no QTLs for salt tolerance was overlapped with the currently detected QTLs in this study. The two the overlapped QTLs whose associated traits are highly related with performance of root phenotype suggests that there may be some overlapped pathway between salt tolerance and root performance.” I am not a native English-speaking person, but I think the manuscript needs a general revision. I copy these sentences as an example, since they are grammarly incorrect. I found many others that are difficult to understand, and that I think someone could re-write in an easier way for readers.

-> Thanks for your kind suggestion. We rewrote those sentences with aid of English editing service

Line 391: we utilized a database

-> We corrected it.

Line 394: “The haplotype analysis for these two genes with the database of previously reported GWAS for salt tolerance showed significant difference among the phenotypic performance of haplotypes of Os04g0481600 and Os04g0493000, providing additional indirect evidence for the possible role in salt tolerance”. Did you mean that? “The haplotype analysis for these two genes, performed with the database of a previously reported GWAS for salt tolerance, showed significant differences for the phenotypic performance under salt conditions of rice cultivars carrying different haplotypes of Os04g0481600 and Os04g0493000, therefore providing additional indirect evidence for their possible role in salt tolerance.”

-> The suggested sentence is much better to explain what we wanted to thus we edited it as the reviewer’s suggestion

Line 412: Please correct “long root length”

-> We corrected.

Line 418: The sentence “among the 44 specifically expressed genes in salt tolerant varieties under the salt treatment” it’s not true. These 44 genes were differentially expressed (up- or down-regulated) under the salt treatment as compared to control conditions in two varieties, a tolerant cultivar and a sensitive one. There are no data about the expression of these genes in a panel of rice varieties. Furthermore, in Line 422, authors wrote “Even though the expressions were downregulated in the tolerant variety under the salt treatment”.

-> We made mistake in writing this sentence. We edited it as “among the 44 specifically expressed genes against salt treatment”

Line 419: Os04g0485300 encodes a glucose-6-phosphate 1-dehydrogenase (G6PDH)…

-> We edited it.

Line 445: “maintaining cell redox balance in rice suspension cultures”

-> We edited it.

Line 446: I suggest to delete these redundant sentences: “Plant GRPs are characterized by a high content of glycine (20-70%). GRP was first discovered in petunia gene.”

-> We deleted them.

Lines 431 and 455 repeat the same information.

-> We deleted the contents in line 455

Lines 432 and 463 repeat the same information.

-> We deleted the contents in line 463

Line 466: “Through a genome-wide association study, we detected three QTLs for salt tolerance in rice seedlings. Bioinformatic analyses performed with …

-> We think this sentence is OK.

Reviewer 3 Report

I agree accept  this manuscipt  in present form

Author Response

Thank you for your review.